# Eutrophication will increase methane emissions from lakes and impoundments during the 21st century

Jake J. Beaulieu [1], Tonya DelSontro [2,4] & John A. Downing [3]

Lakes and impoundments are an important source of methane ($CH_4$), a potent greenhouse gas, to the atmosphere. A recent analysis shows aquatic productivity (i.e., eutrophication) is an important driver of $CH_4$ emissions from lentic waters. Considering that aquatic productivity will increase over the next century due to climate change and a growing human population, a concomitant increase in aquatic $CH_4$ emissions may occur. We simulate the eutrophication of lentic waters under scenarios of future nutrient loading to inland waters and show that enhanced eutrophication of lakes and impoundments will substantially increase $CH_4$ emissions from these systems (+30–90%) over the next century. This increased $CH_4$ emission has an atmospheric impact of 1.7–2.6 Pg C-$CO_2$-eq y$^{-1}$, which is equivalent to 18–33% of annual $CO_2$ emissions from burning fossil fuels. Thus, it is not only important to limit eutrophication to preserve fragile water supplies, but also to avoid acceleration of climate change.

[1] United States Environmental Protection Agency, Office of Research and Development, Cincinnati, OH 45268, USA. [2] Groupe de Recherche Interuniversitaire en Limnologie (GRIL), Département des Sciences Biologiques, Université du Québec à Montréal, Case Postale 8888, Succ. Centre-Ville, Montreal, QC H3C 3P8, Canada. [3] Minnesota Sea Grant and Large Lakes Observatory, University of Minnesota, Duluth, MN 55812, USA. [4] Present address: Department F.-A. Forel for Environmental and Aquatic Science, University of Geneva, Geneva 1211, Switzerland. These authors contributed equally: Jake J. Beaulieu, Tonya DelSontro, John A. Downing. Correspondence and requests for materials should be addressed to J.J.B. (email: Beaulieu.Jake@epa.gov)

The importance of lakes and rivers in the global carbon cycle is well established[1–3]. Local, regional, and global emission estimates of the carbonic greenhouse gases (GHGs), methane ($CH_4$) and carbon dioxide ($CO_2$), are reported regularly[1,2,4,5] with estimates of nitrous oxide ($N_2O$), another potent GHG produced in aquatic systems, also occasionally reported[6]. One of the most recent estimates of global GHG emissions from lakes and impoundments[7] found that while absolute emission of $CO_2$ is 5–10 times more than that of $CH_4$ and $N_2O$ (in Tg of carbon (C) or nitrogen (N) per year), about 72% of the climatic impact of GHG emissions (in $CO_2$-equivalents) from lakes and impounded waters is due to $CH_4$. This is because $CH_4$ is up to 34-times more potent as a GHG than $CO_2$ and is responsible for approximately 20% of the overall additional atmospheric radiative forcing observed since 1750[8].

This recent finding regarding the importance of aquatic $CH_4$ emissions[7] contradicts earlier reports that $CH_4$ and $CO_2$ contributed equally to the global warming potential of GHG emissions from lakes and impoundments[2,4,6]. These previous estimates, however, were based on a simplistic upscaling method of extrapolating average observed emission rates to global lake and impoundment surface area without regard for driving mechanisms. The use of this type of upscaling, rather than a process-based approach, may be the cause for the large uncertainties surrounding global aquatic $CH_4$ emissions[9].

While $CH_4$ emission rates are known to be controlled by a wide range of factors including lake depth[10] and sedimentation rates[11], to name a few, incorporation of drivers into approaches for estimating $CH_4$ emissions has been limited by the lack of world-wide data on these factors. Remote sensing approaches are beginning to fill data gaps, however, and global datasets are now available for lake size and productivity, two important drivers of $CH_4$ emissions[12]. DelSontro et al.[7] used these global datasets, along with >8000 GHG flux measurements, and modeled emission rates as a function of system productivity and lake size. They found, as did a smaller study of impoundments[13], that not only was $CH_4$ the most important GHG emitted from aquatic systems in terms of climate impact, but that it rises exponentially with lake and impoundment chlorophyll $a$ (chla) concentration, a proxy for productivity[7]. This is consistent with multiple lake studies showing that $CH_4$ emissions positively correlate with productivity variables such as total phosphorus (TP) and chla[5,10,13–16]. Ultimately, these relationships reflect the link between an increase in organic substrate and enhanced rates of methanogenesis in productive aquatic systems[14,17,18]. This is an important finding because the productivity of inland waters is projected to increase in the coming decades.

Three distinct mechanisms are expected to induce increases in aquatic productivity over the next century (Table 1, Supplementary Table 1). First, increased human populations (+37% by 2050, +50% by 2100[19]) will augment the release of sewage and agricultural fertilizers to inland waters by an estimated 1.23×–1.97× (e.g., increase by up to a factor of 1.97, expressed as 1.97× hereafter) by 2050 and 1.41×–3.19× by 2100 [20–22]. On a global scale, the spatial distribution of agricultural nutrient use correlates with that of surface water[23], presumably because sustained crop growth requires both nutrients and ample water; therefore, increased global nutrient use is likely to affect inland surface waters. Second, increased storms and runoff will enhance nutrient losses from land by a mean 1.14× over the same period[24,25], further increasing nutrient delivery to inland waters. Third, warming surface waters will increase global aquatic primary production by ~1.30× by 2100 [26]. These increases in eutrophication and consequent $CH_4$ production will be augmented by the 1.10× net global expansion in lakes and impoundment spatial extent[2,26,27] expected over the 21st century,

resulting in more $CH_4$-emitting surface waters. A further increase we cannot assess is enhanced nutrient concentrations in warm regions resulting from increased evaporation, which may be regionally important[26,28], but has not been estimated globally. Projected increases in lake eutrophication are supported by a recent report that the fraction of lakes in the US that are oligotrophic decreased from 25% of all surveyed lakes to 7% over just a 5-year period (2007–2012)[29]. Depending on the trajectory of human population growth and changes in climate and weather, multiplying nutrient effects by exacerbating factors (Table 1, Supplementary Table 1) shows that productivity of lakes and impoundments will likely increase 1.37×–3.10× by 2050 and 2.17×–4.91× by 2100.

Here, we use the relationship between $CH_4$ emission rates and chla reported in DelSontro et al.[7] (Fig. 1) to predict the effect of increased eutrophication of the earth's lakes and impoundments on $CH_4$ emissions. We simulated four levels of increased TP concentration (1.5×, 2×, 2.5×, and 3× that of current levels) that are conservatively within the magnitude of increase predicted by diverse authors and models through 2100 (Table 1). We further simulated a future where improved nutrient management results in TP concentrations 0.75× that of current levels. We calculated the corresponding changes in chla (0.8×, 1.3×, 1.7×, 2.0×, and 2.2× that of current levels) using a non-linear TP—chla relationship derived from published data[30] (Supplementary Figure 1). These relationships were applied to the global lake and impoundment surface area reported in Downing et al.[31]. The current global distribution of chla in lakes and impoundments was estimated using satellite-based measurements of chla in 80,000 lakes around the world[32]. Our simulation results indicate that enhanced eutrophication will increase $CH_4$ emissions from lakes and impoundments by 30–90% over the next century.

## Results

**Enhanced $CH_4$ emissions due to future productivity increases.** Our model indicates that future eutrophication of the world's lakes and impoundments will increase diffusive, ebullitive, and total $CH_4$ emissions (Fig. 2). Assuming a 3× increase in TP concentrations, diffusive emissions may increase from current

**Table 1 Changes driving expanded eutrophication and rising $CH_4$ emissions from lakes and impoundments during the 21st century**

| | Citation | Change to 2050 | | Change to 2100 | |
|---|---|---|---|---|---|
| | | N | P | N | P |
| **Changes in fertilizer production and nutrient runoff**[a] | | | | | |
| Diverse population | [19–21] | 1.23×[b] | | 1.41×[b] | |
| driven increases in | 38 | 1.19× | 1.50× | | |
| nutrients from | 37 | 2.70×[c] | 2.44×[c] | | |
| crop-livestock | 22 | | 1.97×[b] | | 3.19×[b] |
| systems | 19 | 1.37× | | 1.50× | |
| **Changes that will augment the effect of changes in fertilizer production and nutrient runoff**[a] | | | | | |
| Climate-driven nutrient runoff | [24–26] | 1.10× | | 1.14× | |
| Water temperature | 26 | | | 1.30× | |
| Lake area | 26 | 1.05× | | 1.10× | |
| **Aggregate effect on eutrophication**[a,d] | | | | | |
| | | 1.37×–3.10× | | 2.17×–4.91× | |

[a]See Supplementary Table 1 for additional details
[b]Nutrient leaching
[c]Fertilizer use
[d]Aggregate effect calculated as change in fertilizer production and nutrient runoff multiplied by the sum of the augmentation factors for each time period. Ranges represent estimates using the minimum and maximum changes in fertilizer production and nutrient runoff

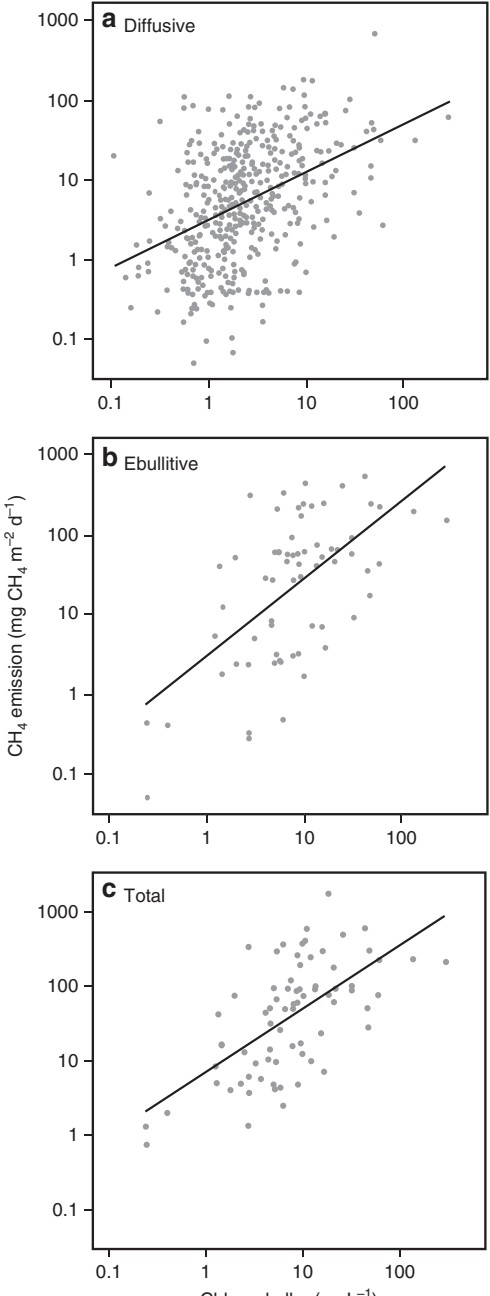

**Fig. 1** Relationships between $CH_4$ flux rates and productivity measured as the concentration of chlorophyll $a$ (chla) in the water column of lakes and impoundments. Solid lines are regression relationships. **a** Diffusive flux is the flux rate from the water to the atmosphere driven by super-saturation in the water column. **b** Ebullitive flux is that emitted by the formation of bubbles transported to the surface. **c** Total flux is the sum of both diffusive and ebullitive flux. Note: panel **a** shows only the relationship between diffusive $CH_4$ flux and chla concentration, whereas the predictive model for diffusive flux includes both a lake size and chla effect (Table 3)

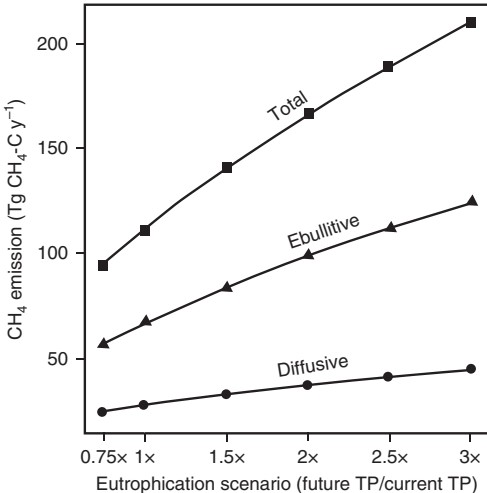

**Fig. 2** Simulated future global emissions of $CH_4$ calculated at multiple levels of increased eutrophication driven by accelerated TP loading to lakes and impoundments (see Methods). The scenarios are based on a review of the published literature and reflect future TP concentrations that are 0.75, 1.5, 2, 2.5, and 3× greater (i.e., 0.75×, 1.5×, 2×, 2.5×, 3×) than current levels (1×), as indicated on x-axis. Simulated TP concentration was converted to chlorophyll $a$ (chla) using published data (see Methods and Supplementary Figure 1) resulting in future chla concentrations that are 0.8, 1.3, 1.7, 2.0, and 2.2× that of current levels. $CH_4$ emissions were predicted from lake surface area and/or chla

levels (28 Tg C-$CH_4$ y$^{-1}$) up to 45 Tg $CH_4$-C y$^{-1}$, representing a 60% increase in emissions. Ebullition and total (diffusive + ebullitive) $CH_4$ emission rates are more sensitive to enhanced eutrophication and will nearly double (mean increase of 1.86×) in a future environment with 3× greater nutrient loading to lakes and impoundments (Supplementary Table 2).

At current productivity levels, we find that total $CH_4$ emissions (112 Tg C-$CH_4$ y$^{-1}$) are approximately 25% higher than the latest global estimates for lakes and impoundments (88.7 Tg C-$CH_4$ y$^{-1}$)[9], which was based on a synthesis of measurements from mostly northern systems (>50°N) that used the traditional upscaling approach. Under a scenario of a tripling of nutrient loading, we could find the aggregate of global lakes and impoundments emitting more than double the current global estimate. More concerning yet is that even at the smallest increase in TP suggested by the literature (1.5×) we would see total global lake $CH_4$ emissions rise to 141 Tg C-$CH_4$ y$^{-1}$, equivalent to global emissions from wetlands, which is the current single largest source of atmospheric $CH_4$ (139 Tg C-$CH_4$ y$^{-1}$)[9]. A tripling of nutrient loading could result in total lake $CH_4$ emissions being 1.5× higher than the current wetland emission estimate. It is possible, though unlikely, that improved nutrient management practices could reverse the eutrophication of lakes and impoundments, thereby reducing $CH_4$ emissions. For example, a 25% reduction in global TP loading (0.75×) could prevent the emissions of 17 Tg $CH_4$-C y$^{-1}$ (Fig. 2, Supplementary Table 2). While a net reduction in nutrient loading is unlikely to occur on a global scale due to increasing population, it may be realized regionally or for individual water bodies and could be included in the cost/benefit analysis of nutrient management programs[33].

Our literature review indicates that global TP loading of inland waters will increase over the 21st century, stimulating aquatic productivity and $CH_4$ emissions. We find that these changes could result in total lake $CH_4$ emissions having an atmospheric impact equivalent to 18–33% of that from current fossil fuel $CO_2$ emissions (Table 2). In fact, the net increase in $CH_4$ emissions due to projected nutrient loading alone would have an atmospheric effect more than increased $CO_2$ emissions from land use change (i.e., conversion of forests to agriculture) or up to half of the global oceanic and land carbon sinks (Table 2[3,34]). The projected $CH_4$ emission increases reported here should be considered minimal because they do not account for the synergistic interaction between nutrients and water temperature on $CH_4$ production rates[39], future increases in impounded area[27], or the effect of eutrophication on $CH_4$ emissions from streams,

**Table 2 Projected increases in $CH_4$ emissions from lakes and impounds as compared to major global $CO_2$ sources and sinks**

| Source/sink | Pg C-$CO_2$ $y^{-1\ a}$ | Fraction of source/sink represented by future $CH_4$ emissions from lakes and impounds | |
| --- | --- | --- | --- |
| | | Due to total future $CH_4$ emissions[b] | Due to future change in $CH_4$ emissions[b] |
| Fossil fuels burning | 7.8 and 9.3 | 0.18–0.33× | 0.03–0.15× |
| Land use change | 1.1 and 1.0 | 1.54–2.61× | 0.27–1.22× |
| Oceanic sequestration | −2.3 and −2.6 | 0.65–1.13× | 0.12–0.53× |
| Land sequestration | −2.6 and −3.1 | 0.55–1.0× | 0.10–0.47× |

[a]Values from Ciais et al.[3] and Le Quéré et al.[34]
[b]Minimum values calculated as the projected total $CH_4$ emissions under the minimum future TP loading scenario (1.5× scenario; Supplementary Table 2) divided by the maximum source/sink estimate. Maximum values calculated as the maximum projected total $CH_4$ emissions under the maximum future TP loading scenario (3× scenario; Supplementary Table 2) divided by the minimum source/sink estimate

**Table 3 Statistical models used to predict methane ($CH_4$) emission rates**

| Model[a] | Coefficient of determination ($r^2$) | p | n |
| --- | --- | --- | --- |
| $\log_{10}$(diffusive $CH_4$ emission rate + 1) = −0.167*$\log_{10}$(SA) + 0.530*$\log_{10}$(chla) + 0.098*$\log_{10}$(SA)*$\log_{10}$(chla) + 0.705 | 0.29 | <0.001 | 423 |
| $\log_{10}$(ebullitive $CH_4$ emission rate + 1) = 0.758*$\log_{10}$(chla) + 0.752 | 0.32 | <0.001 | 65 |
| $\log_{10}$(total $CH_4$ emission rate) = 0.778*$\log_{10}$(chla) + 0.940 | 0.38 | <0.001 | 74 |

[a]Units for emission rates, SA, and chla are mg $CH_4$-C $m^{-2}$ $d^{-1}$, $km^2$, and µg $L^{-1}$, respectively.
All terms in the models were significant at the $p < 0.001$ level. See DelSontro et al.[7] for details

rivers, small ponds, and wetlands. Eutrophication is considered to be one of the world's most pressing environmental issues[35] and, if it continues, will exacerbate global climate change.

## Methods

**Experimental design.** We simulated future TP loadings to lakes and impoundments that are 0.75×, 1.5×, 2×, 2.5×, or 3× that of current loading, a conservative set of scenarios relative to our literature review which indicated that TP may increase nearly 5× relative to current levels by 2100 (Table 1, Supplementary Table 1 [20,22,24,][36]). Changes in TP concentration were translated to changes in chlorophyll a (chla) concentration using the non-linear TP—chla relationship derived from the data in McCauley et al.[30] (Supplementary Figure 1) and an estimate of the current global chla distribution in lakes and impoundments. The global distribution of chla in lakes and impoundments was derived from satellite-based chla measurements of 80,000 lakes around the world[32]. We combined these data with an estimate of the size distribution of lakes and impoundments[31] to generate a joint lake-size by productivity distribution for the worlds lakes and impoundments. Although several global lake-size distributions are available in the literature, all of them offer very similar conclusions. The data were aggregated into twenty 5 µg $L^{-1}$ chla bins and nine lake-size bins ranging from 0.001 to >100,000 $km^2$.

These TP loading scenarios (0.75×, 1.5×, 2×, 2.5×, or 3×) resulted in future chla concentrations that are, on average, 0.8×, 1.3×, 1.7×, 2.0×, and 2.2× that of the current global chla distribution. We propagated each of these chla distributions across the joint lake-size by chla distribution, resulting in five new distributions reflecting differing levels of eutrophication (Supplementary Data 1).

We calculated global $CH_4$ emissions for each scenario using statistical models relating diffusive, ebullitive, and total $CH_4$ emission rates to lake size and/or water column chla content (Table 3). First, we calculated the mean areal emission rate (mg $CH_4$-C $m^{-2}$ $d^{-1}$) for each lake-size by chla bin. Next, we multiplied the predicted areal emission rate for each bin by the total water body surface area corresponding to the bin. Finally, we summed the emissions across bins and scaled the result to an annual emission estimate for each emission mechanism (i.e., diffusive, ebullitive, and total emissions).

The range of TP and chla concentrations in the most extreme 3× simulation (11–1669 µP $L^{-1}$ and 5.7–216 µg $L^{-1}$, respectively) were well within the range of values included in the literature used to parameterize the statistical models[7] (Fig. 1). Statistical uncertainty in the model predictions was propagated through the calculations and is presented as 95% confidence intervals in Supplementary Table 2 and Supplementary Data 1.

The statistical models used to predict $CH_4$ emission rates do not include an effect for water body origin (i.e., natural vs constructed) because of insufficient data coverage across the range of covariates for both system types. There is little evidence in the literature that suggests $CH_4$ emission rates differ between natural lakes and impoundments, after system size and productivity are accounted for. Furthermore, impoundments constitute a small fraction of total inland water surface area[31], therefore any systematic difference in the response of $CH_4$ emission

rate to size or productivity between natural lakes and impoundments will likely have little effect on cumulative global $CH_4$ emissions from lentic waters.

**Statistical analysis.** The models used to upscale lake and impoundment methane ($CH_4$) emissions are presented in DelSontro et al.[7] and are reproduced in Table 3. The models predict $CH_4$ emission rates (mg $CH_4$-C $m^{-2}$ $d^{-1}$) from lake size ($km^2$) and/or chla (µg $L^{-1}$).

## Data availability

All data used to generate Figure 1 and the models presented in Table 3 are available through figshare, an open-source data repository (https://doi.org/10.6084/m9.figshare.5220001). Joint lake-size and chla distribution tables for all modeled scenarios are available as Supplementary Data 1.

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

## Acknowledgements

This work was supported by the Minnesota Sea Grant College Program and by the Industrial Research Chair in Carbon Biogeochemistry in Boreal Aquatic Systems (CarBBAS), co-funded by the Natural Sciences and Engineering Research Council of Canada (NSERC) and Hydro-Québec. The joint lake-size by chla distribution, $CH_4$ emission rates, and corresponding chla values are available at the figshare data repository (https://figshare.com/s/0a39281088d644a1d925). This document has been reviewed in accordance with the U.S. Environmental Protection Agency policy and approved for publication. The views expressed in this paper are those of the authors and do not necessarily reflect the views or policies of the U.S. Environmental Protection Agency.

## Author contributions

T.D., J.J.B., and J.A.D. identified the research questions, designed the study approach, and wrote the manuscript. J.A.D. identified the upscaling approach and compiled data on future nutrient loading to lakes and impoundments. J.J.B. wrote the code to implement the scenarios.

## Additional information

**Competing interests:** The authors declare no competing interests.

