## [Peer Review File · Nature Communications]

Reviewers' comments:

Reviewer #1 (Remarks to the Author):

This is a well-written paper and presents a simple, but interesting spatiotemporal extrapolation of our current understanding of drivers of methane emissions from lakes and reservoirs. My only major concern with this paper is how ebullitive CH₄ release was extrapolated for the global lake area. Much evidence has shown that sites with high ebullition usually have shallower water columns. It seems to me that there must be an effect of lake size and/or an interaction between lake size and productivity for predicting lake-wide ebullitive flux of CH₄. In a large, deep lake diffusive emissions will dominate and the ebullitive response to enhanced productivity should not be a major driver of the whole-lake response, but with the authors' approach the lake emissions increase dramatically due to changes in diffusive and ebullitive emissions. I understand that the 144 observations used in the del Sontro et al. (2018) paper didn't reveal a statistical relationship between these parameters and ebullitive CH₄ emission rates, but if the authors are truly proponents of a "process-based approach" (Line 42) they should think about how to incorporate these physical/geomorphic effects.

Reviewer #2 (Remarks to the Author):

Eutrophication will increase methane emissions from lakes and impoundments during the 21st century

The article is well written and has a clear quantitative message: 'Eutrophication driven increase in methane emission amounts up to 30-90% during the 21st century'. The authors used an interesting and well-explained upscaling method to come to this number. I found the methodology original and see great potential for this paper to be published in Nature Communication. Still, I see room for improvement. I have remarks regarding some of the assumptions made and regarding the way the available data was included in the upscaling model.

Assumptions

It is not clear to me how the authors translated the listed increases in nutrient (often N) loading into a potential 3-fold increase in TP concentration in lakes. A better explanation about how the range of 1.5 -3-fold increase was calculated is needed. Loading and concentration are not likely 1:1 related see Eg. Jeppesen et al. 2011 who found an increase in P-runoff but no increase in lake concentration.

Is a 3-fold increase realistic for a considerable share of the inland waters? Perhaps for the most oligotrophic waters where a small increase may cause a fold-increase in TP, but this will have little impact on CH₄ emissions as you show in your figures. See also my remarks below.

In addition, the anoxia amplifier you use can be criticized. The data presented in Jenny et al. clearly shows that the expansion of hypoxia in inland waters has leveled off. Using the slope of the years 1945-2000 likely results in a strong over estimation of the expansion of anoxic waters. I suggest to remove this amplifier from the model as it undermines the credibility of the model.

Model

Your estimated increase in CH₄ emissions is based on world-wide average in chl-a

concentration. You start of using a global average chl-a concentration, which disregards that the majority of lakes have a chl-a concentration below this average. As an increase in (TP as well as in) chl-a has a proportionally lower absolute effect on CH₄ emissions this leads to an overestimation of the effect of enhanced

eutrophication on CH₄ emissions. The global distribution of chl-a is known (published in Sayer). I suggest you use this distribution in your model to obtain a more realistic (lower) estimate of emission increase. This should be possible at least for model (3) where lake size is not included.

In addition to this: Inevitably one needs to make assumptions when applying an upscaling approach like yours. And all come with their uncertainties which propagate. Can you elaborate on the uncertainty of your final outcome?

Other remarks

I suggest to include a brief statement about the methodology in the abstract

Line 58. What do you mean with 'distribution'?

Line 68 Do you mean the areal extend of inland waters?

Line 70 This ref. is about nitrogen, whereas remainder of this m.s. deals mostly with P. We suggest to include earlier work in Denmark:

Jeppesen E, Kronvang B, Meerhoff M, Søndergaard M, Hansen KM, Andersen HE, Lauridsen TL, Beklioglu M, Ozen A, Olesen JE. 2009. Climate change effects on runoff, catchment phosphorus loading and lake ecological state, and potential adaptations. *Journal of Environmental Quality*. 38:1930-1941.

Jeppesen E, Kronvang B, Olesen J, Audet J, Søndergaard M, Hoffmann C, Andersen H, Lauridsen T, Liboriussen L, Larsen S, Beklioglu M, Meerhoff M, Özen A, Özkan K. 2011. Climate change effects on nitrogen loading from cultivated catchments in Europe: implications for nitrogen retention, ecological state of lakes and adaptation. *Hydrobiologia*. 663:1-21.

Line 76 this study deals with lakes > 5 m deep which have a different dynamic than shallow systems. Some argumentation on why this may also be applicable to shallower systems where most CH₄ emission (especially ebullition) takes place is needed.

In addition: How were these percentages derived? The paper itself does not seem to mention them and certain assumptions seem to be needed to derive them from the published data.

Line 116 also wetland emissions are likely to increase

Line 124 Tranvik et al. Points out to much of the Earth will experience declines in lake abundance (not an increase).

Line 166-168. how well do these models predict the rates? Please provide the statistics.

Table 1: the lay-out of this table could be improved. Eg. It is not fully clear to which lines the "Population driven expansion of croplivestock systems" is applicable

Line 286 More explanation about how this was done would be very helpful.

Line 294. The Note was a bit confusing to me as the overall diffusive emissions are not shown?

Table S1 "This is likely an underestimate because inland production is driven by P that moves with N but it ignores land-water processing in inland waters and nutrient retention." This sentence was not clear to me.

Sinha et al. 'nutrients' should read nitrogen.

Reviewer #3 (Remarks to the Author):

The manuscript “Eutrophication will increase methane emission from lakes and impoundments during the 21st century” by JJ Beaulieu, T DelSontro and JA Downing presents a prediction on the development of methane emission from freshwater lakes and impoundments in this century. The interesting approach is important for our understanding of different impacts on global climate. It gives important information to the management of lakes and reservoirs now and in future.

The authors hypothesize that methane emission will increase in coming decades due to eutrophication of inland waters. A huge data-set on methane emissions from lakes and impoundments was compiled and published recently by the same authors (DelSontro et al., L&O Lett 2018).

Three main facts were identified as drivers of eutrophication and by this of CH₄-emissions from lakes and impoundments. This is first the increasing human population which is set in relation to the use of fertilizers, production of sewage and the transport of both into freshwaters. This statement is not clearly formulated (definition of release: What is meant: used in agriculture or released into aquatic environments) and not further discussed. It stays unclear e.g., if societies will undertake measures to minimize pollution of aquatic habitats. In Europe, the water framework directive is dedicated to act in this direction and improve the water quality of freshwaters. Secondly, nutrient runoff by storms and water runoffs is described as eutrophication forcing. Well referenced numbers on forecasted increases in nutrient contents, in hypoxia, in trophic level decrease etc. are given. All of them together arouse the impression of mainly degraded aquatic systems in the near future. That brings up the question about the changes in methane emission when the systems are already degraded and will change from eutrophic to hyper-eutrophic level. And thirdly, the increasing water temperature due to global climate change is mentioned. The authors should have mentioned here how an increasing water demand by human beings and an increased evaporation by increasing temperatures fit to the projected increase (10%) in freshwater surface area.

An explanation of the processes by which an increasing productivity leads to higher methane emission would help to follow the different drivers. Is it just the increasing amount of sedimenting carbon, or is it the increase in hypoxia? Which role plays methane oxidation in mitigating emission? I recommend to consider recent publications estimating the different impacts of climate warming on the antagonistic microbial processes of methane production and –oxidation (e.g., Sepulveda-Jauregui et al., 2018, STOTEN).

The two mentioned pathways of methane release from the waters to the atmosphere need a more detailed discussion. Diffusion depends mainly on the methane concentration in the top water layer. How does the surface CH₄-concentration increase as result of eutrophication? The surface water receives oxygen from the atmosphere and remains oxic throughout the year. What drives ebullition?

Beside lake surface area, also water depth and mixing regimes should be considered.

Emissions from Impoundments differ substantially on a global scale.

In summary, the manuscript summarizes a number of paper concerning TP and Chla estimations and forecasts, the basis of emission data is the mentioned data-set published by the authors.

The manuscript is of general interest to those working in this field of science, but results will not stimulate new thinking in the field. The main merit is the compilation of former published data. The interpretation is based on simplistic upscaling over coming 80 years without regard for interactive natural processes and societal activnesses.

Reviewer/Location/Revised location	Comment	Done
1/General	My only major concern with this paper is how ebullitive CH₄ release was extrapolated for the global lake area. Much evidence has shown that sites with high ebullition usually have shallower water columns. It seems to me that there must be an effect of lake size and/or an interaction between lake size and productivity for predicting lake-wide ebullitive flux of CH₄. In a large, deep lake diffusive emissions will dominate and the ebullitive response to enhanced productivity should not be a major driver of the whole-lake response, but with the authors' approach the lake emissions increase dramatically due to changes in diffusive and ebullitive emissions. I understand that the 144 observations used in the del Sontro et al. (2018) paper didn't reveal a statistical relationship between these parameters and ebullitive CH₄ emission rates, but if the authors are truly proponents of a "process-based approach" (Line 42) they should think about how to incorporate these physical/geomorphic effects.	Many factors have been shown to correlate with emission rates, but only those factors for which the global distribution is known can be used for global upscaling exercises. Water depth may be an important predictor, but the global lake depth distribution is not known, therefore this variable cannot be used for global upscaling. We are also uncertain if system-scale emission rates correlate with system-scale depth estimates. While several studies have shown that emission rates vary with depth within a water body, mean or maximum lake depth has not been shown to be a strong predictor of emission rates among systems. In fact, relatively small areas of intense methanogenic activity can produce relatively high emissions from an otherwise deep waterbody (DelSontro, T. et al. Spatial Heterogeneity of Methane Ebullition in a Large Tropical Reservoir. Environ. Sci. Technol. 45, 9866-9873, (2011)). Please see the expanded Introduction for a clarifying discussion (lines 46-51 in revised MS).
2	It is not clear to me how the authors translated the listed increases in nutrient (often N) loading into a potential 3-fold increase in TP concentration in lakes. A better explanation about how the range of 1.5 -3-fold increase was calculated is needed. Loading and concentration are not likely 1:1 related see Eg. Jeppesen et al. 2011 who found an increase in P-runoff but no increase in lake concentration.	We have greatly expanded Table S1, including the table caption, to more clearly state our reasoning in making these projections. We also reformatted table 1, table S1 and the text to provide greater clarity on the source and use of the 1.5-3-fold increase.
2	Is a 3-fold increase realistic for a considerable share of the inland waters? Perhaps for the most oligotrophic waters where a small increase may cause a fold-increase in TP, but this will have little impact on CH₄ emissions as you show in your figures. See also me remarks below.	Please see new text in Table S1 caption.

2	In addition, the anoxia amplifier you use can be criticized. The data presented in Jenny et al. clearly shows that the expansion of hypoxia in inland waters has leveled off. Using the slope of the years 1945-2000 likely results in a strong over estimation of the expansion of anoxic waters. I suggest to remove this amplifier from the model as it undermines the credibility of the model.	We have removed the hypoxia amplifier from the manuscript.
2	Your estimated increase in CH4 emissions is based on world-wide average in chl-a concentration. You start of using a global average chl-a concentration, which disregards that the majority of lakes have a chl-a concentration below this average. As an increase in (TP as well as in) chl-a has a proportionally lower absolute effect on CH4 emissions this leads to an overestimation of the effect of enhanced eutrophication on CH4 emissions. The global distribution of chl-a is known (published in Sayer). I suggest you use this distribution in your model to obtain a more realistic (lower) estimate of emission increase. This should be possible at least for model (3) where lake size is not included.	We agree with the reviewer's suggestion and have implemented it for all emission mechanisms (i.e. diffusion, ebullition, and total). As the reviewer suggested, the revised approach predicts lower emissions under elevated nutrient loading, but the difference was quite small. For example, predicted emissions under a 3-fold increase in TP are <1% lower when calculated using the revised approach (210 Tg CH4-C y-1), as compared to the original approach (211 Tg CH4-C y-1). This finding is a bit surprising given that the two approaches entail very different upscaling methodologies and points to the robustness of the productivity ~ CH4 relationship at the global scale. The numbers have been updated throughout the paper. See the Method section for a detailed description of the revised approach.
2	In addition to this: Inevitably one needs to make assumptions when applying an upscaling approach like yours. And all come with their uncertainties which propagate. Can you elaborate on the uncertainty of your final outcome?	One source of uncertainty is what the future environment will look like, which is why we ran four different future scenarios and present the range of possible outcomes in Table 2. Another source of uncertainty is error in the underlying statistical model. In the revised paper we propagate this error through the calculations and include 95% confidence intervals with the emission estimates (Table S2).
2/Abstract	I suggest to include a brief statement about the methodology in the abstract	We have added a brief statement to abstract.
2/Line 58	What do you mean with 'distribution'?	Replaced with 'dissolved CH4 concentration'
2/Line 68	Do you mean the areal extend of inland waters?	Rewritten for clarity.

2/Line 70/Line 66	This ref. is about nitrogen, whereas remainder of this m.s. deals mostly with P. We suggest to include earlier work in Denmark: Jeppesen E, Kronvang B, Meerhoff M, Søndergaard M, Hansen KM, Andersen HE, Lauridsen TL, Beklioglu M, Ozen A, Olesen JE. 2009. Climate change effects on runoff, catchment phosphorus loading and lake ecological state, and potential adaptations. Journal of Environmental Quality. 38:1930-1941. Jeppesen E, Kronvang B, Olesen J, Audet J, Søndergaard M, Hoffmann C, Andersen H, Lauridsen T, Liboriussen L, Larsen S, Beklioglu M, Meerhoff M, Özen A, Özkan K. 2011. Climate change effects on nitrogen loading from cultivated catchments in Europe: implications for nitrogen retention, ecological state of lakes and adaptation. Hydrobiologia. 663:1-21.	Thank you for bringing these papers to our attention. We have added Jeppesen et al 2009 to Tables 1 and S1. We considered papers that predicted future changes in either N or P, as indicators of future eutrophication. See Table S1 caption for new text justifying this approach.
2/Line 76/Line 75	this study deals with lakes > 5 m deep which have a different dynamic than shallow systems. Some argumentation on why this may also be applicable to shallower systems where most CH4 emission (especially ebullition) takes place is needed. In addition: How were these percentages derived? The paper itself does not seem to mention them and certain assumptions seem to be needed to derive them from the published data.	The reference to lakes >5m deep pertains to the cited hypoxia study. We have removed this study and any reference to changes in anoxia from the MS. We have reformatted Table 1, S1, and text to provide greater transparency on how the percentages (now expressed as factors instead of percentage increases) were derived and used.
2/Line 116/Line 153	also wetland emissions are likely to increase	We added this to the list.
2/Line 124/Line 86	Tranvik et al. Points out to much of the Earth will experience declines in lake abundance (not an increase).	Tranvik et al., which was coauthored by J. Downing, an author of the current work, reports a net increase in global surface area, despite reductions in some regions. We now state "net global extent" in the paper.
2/Lines 166-168/Table 3	how well do these models predict the rates? Please provide the statistics.	This information is now reported in a new table (Table 3).
2/Table 1	he lay-out of this table could be improved. Eg. It is not fully clear to which lines the "Population driven expansion of croplivestock systems" is applicable	The table has been reformatted for improved readability.

2/Line 286/Line 324	More explanation about how this was done would be very helpful.	Tables 1 and S1 have been reorganized for clarity. Additionally, Table S1 now provides additional information on how the numbers were extracted from the literature.
2/Line 294/Line 342	The Note was a bit confusing to me as the overall diffusive emissions are not shown?	Rephrased note to improve clarity.
2/Table S1	“This is likely an underestimate because inland production is driven by P that moves with N but it ignores land-water processing in inland waters and nutrient retention.” This sentence was not clear to me.	This has now been revised in the first “estimator and notes” cell of Table S1
2/Lines 70-71/Line 83	Sinha et al. ‘nutrients’ should read nitrogen.	We added a reference to Jeppesen et al. 2009 which reports increased P export due to climate change. We therefore retain ‘nutrients’ in this sentence.
3/	This is first the increasing human population which is set in relation to the use of fertilizers, production of sewage and the transport of both into freshwaters. This statement is not clearly formulated (definition of release: What is meant: used in agriculture or released into aquatic environments) and not further discussed.	Added text (line 77, ‘to inland waters’ Line 82 ‘losses from land’ Line 107 “or its surrogates”). Text added throughout Table S1 to clarify links between population growth and eutrophication of inland waters.
3/	It stays unclear e.g., if societies will undertake measures to minimize pollution of aquatic habitats. In Europe, the water framework directive is dedicated to act in this direction and improve the water quality of freshwaters	We now include a scenario whereby a 25% reduction in nutrient loading is achieved through improved nutrient management. See lines 138-143 for a discussion of the results.
3/	That brings up the question about the changes in methane emission when the systems are already degraded and will change from eutrophic to hyper-eutrophic level.	The TP and chl values used in the simulations are within the range of literature values used to parameterize the statistical model. We do not extrapolate beyond the data. See Line 178-182 for this explanation
3/	The authors should have mentioned here how an increasing water demand by human beings and an increased evaporation by increasing temperatures fit to the projected increase (10%) in freshwater surface area.	We have added two citations and a discussion of this effect. See Line 87 and Table S1 caption.

3/	An explanation of the processes by which an increasing productivity leads to higher methane emission would help to follow the different drivers. Is it just the increasing amount of sedimenting carbon, or is it the increase in hypoxia? Which role plays methane oxidation in mitigating emission? I recommend to consider recent publications estimating the different impacts of climate warming on the antagonistic microbial processes of methane production and –oxidation (e.g., Sepulveda-Jauregui et al., 2018, STOTEN).	Thank you for the suggestion. We now provide a more thorough discussion of the underlying mechanisms, and their interaction with climate-warming, in the revised ‘Productivity drives methane emission rates’ section.
3/	The two mentioned pathways of methane release from the waters to the atmosphere need a more detailed discussion. Diffusion depends mainly on the methane concentration in the top water layer. How does the surface CH ₄ -concentration increase as result of eutrophication? The surface water receives oxygen from the atmosphere and remains oxic throughout the year. What drives ebullition?	We now discuss the two emission mechanisms in the revised ‘Productivity drives methane emission rates’ section.
3/	Beside lake surface area, also water depth and mixing regimes should be considered.	Many factors have been shown to correlate with emission rates, but only those factors whose global distribution is known can be used for global upscaling exercises. Water depth and mixing likely are important, but information on these factors is not available globally. Please see the expanded Introduction for a discussion (lines 46-50).
3/	Emissions from Impoundments differ substantially on a global scale.	Our model assumes no difference in emission rates natural lakes and impoundments, after system size and productivity are accounted for. We now clearly state this assumption and discuss its implications. See lines 183-189 in revised MS.

REVIEWERS' COMMENTS:

Reviewer #1 (Remarks to the Author):

The authors have addressed my previous concerns by adding some additional text. I still view this set of projections to be extremely speculative and not based in our understanding of physiochemical processes, but perhaps this manuscript will spur additional work at local and broader scales to address these gaps.

Reviewer #2 (Remarks to the Author):

I thank the authors for critically reflecting on the issues raised by me and the other reviewers. I find the manuscript improved in clarity and methodology. I think the paper is of high interest to a wide audience of Nature Communication readers. At this point I only have a few additional comments/ suggestions/questions:

Line 167 I suggest to add '20' before 5 μg . [20 bins]

Line 163- 167 I find this approach very interesting and as the authors am surprised be the lack of effect of incorporating chla-a distribution patterns on the total increase in CH₄ emissions. I highly recommend to present the results of the join lake-size by chla distribution table. It is an important step in your calculations and can help the readers to verify your results.

I guess that the reason for the lack of effect of incorporating this more sophisticated approach is because the total areal extend of the (very) low chla lakes is low and therefore the modest increase in CH₄ emissions in these lakes does not affect the total global increase. Is my reasoning correct?

Line 168. I suggest to slightly re-write. The 0.8x etc are a resultant of your non-linear relationship with TP. Here it reads as if it has something to do with the distribution

Line 325. I suggest to remove 'sum of' as it may confuse the reader.

Line 328 I do not fully understand the column headings 'Due to total future CH₄ emissions' and 'Due to future change in Ch₄ emissions'. I expect the left column to include the change (in CH₄ from lakes and impoundments) and therefore to represent a higher fraction of for instance fossil fuel burning than the values in the right column where only the change in CH₄ emission in lakes and impoundments is compared to fossil fuel burning. Spend some time trying to understand this table. I suggest to clarify more.

Table 3. I suggest to give more statistical information (p-values of models as well as the different parameters; n-values).

Reviewer/Location/Revised location	Comment	Response
#2/Line 167/Line 169	suggest to add '20' before 5 µg. [20 bins]	Added 'twenty'
#2/Line 163-167/Lines 159-163	I find this approach very interesting and as the authors am surprised be the lack of effect of incorporating chla-a distribution patterns on the total increase in CH4 emissions. I highly recommend to present the results of the joint lake-size by chla distribution table. It is an important step in your calculations and can help the readers to verify your results. I guess that the reason for the lack of effect of incorporating this more sophisticated approach is because the total areal extend of the (very) low chla lakes is low and therefore the modest increase in CH4 emissions in these lakes does not affect the total global increase. Is my reasoning correct?	The lack of an effect is also related to the range of TP/chla values that were modeled in the original submission vs. the revised manuscript. In the original submission, all scenarios were referenced against a global mean TP concentration (~50 ug/L). In the most extreme scenario (3-fold increase in TP), the maximum modeled TP value was ~150 ug/L. In the revised approach we no longer used a global mean TP and chla, rather we used the full range of chla values reported in Sayers et al (5-100 ug/L), which equates to TP values ranging from (~5-600 ug/L). The range of TP values further increased in the future scenarios. With revised approach, TP values ranged from ~15-1800 ug/L in the most extreme scenario. This broad range of values encompassed the non-linear portions of the chla~TP curve, such that lower TP lakes had a proportionally higher chla increase whereas the higher TP lakes had a proportionally lower increase. Lakes with moderate TP exhibited chla increases similar to those reported in the original submission because the chla ~ TP relationship is approximately linear in this range. Overall, the revised approach made little difference partly because the larger increases in the low TP lakes were compensated for by the smaller increases in the higher TP lakes. Please see new Supplemental Table 3 for the joint lake-size by chla/TP tables.
#2/Line 168/Line 171	I suggest to slightly re-write. The 0.8x etc are a resultant of your non-linear relationship with TP. Here it reads as if it has something to do with the distribution	Done

#2/Line 325/Line 355	I suggest to remove ‘sum of’ as it may confuse the reader.	We now specify “the sum of”
#2/Table 2	I do not fully understand the column headings ‘Due to total future CH4 emissions’ and ‘Due to future change in Ch4 emissions’. I expect the left column to include the change (in CH4 from lakes and impoundments) and therefore to represent a higher fraction of for instance fossil fuel burning than the values in the right column where only the change in CH4 emission in lakes and impoundments is compared to fossil fuel burning. Spend some time trying to understand this table. I suggest to clarify more.	There was an error in row 1 of the table. ‘1.03’ and ‘1.56’ were written when the correct values are ‘0.03’ and ‘0.56’. The table now reads as the reviewer expected.
#2/Table 3	I suggest to give more statistical information (p-values of models as well as the different parameters; n-values).	Done